# Screening, Identification and Growth-Promotion Products of Multifunctional Bacteria in a Chinese Fir Plantation

**Guangyu Zhao [1,2,3], Yihui Wei [1,2,3], Jiaqi Chen [1,2,3], Yuhong Dong [1,2,3], Lingyu Hou [1,2,3] and Ruzhen Jiao [1,2,3,*]**

[1] Research Institute of Forestry, Chinese Academy of Forestry, Beijing 100091, China; m15933021570@163.com (G.Z.); weiyh@caf.ac.cn (Y.W.); cjq030327@163.com (J.C.); dongyh@caf.ac.cn (Y.D.); houlingyu@caf.ac.cn (L.H.)

[2] State Key Laboratory of Tree Genetics and Breeding, Chinese Academy of Forestry, Beijing 100091, China

[3] Key Laboratory of Tree Breeding and Cultivation of State Forestry Administration, Chinese Academy of Forestry, Beijing 100091, China

\* Correspondence: jiaorzh@caf.ac.cn; Tel.: +86-010-6288-9663

**Abstract:** *Purpose*: This research was aimed to screen and identify multifunctional phosphorus-dissolving bacteria of a Chinese fir (*Cunninghamia lanceolata*) plantation and study its phosphorus-dissolving characteristics in order to provide strain resources and a theoretical basis for developing the appropriate bacterial fertilizer of a Chinese fir plantation. *Methods*: First, phosphorus-dissolving bacteria were isolated from the woodland soil of a Chinese fir plantation by Pikovskava inorganic phosphorus medium (PVK). Then, some growth-promoting indicators of primary screening strains were determined, including the capacity of phosphorus-solubilized, nitrogenase activity, 1-aminocyclopropane-1-carboxylic acid (ACC) deaminase activity, production of indole-3-acetic acid (IAA), secretion of iron carrier and so on. Finally, the screening multifunctional phosphorus-dissolving bacteria were identified, which were combined with colony characteristics, physiological and biochemical tests and molecular biotechnology. *Results*: (1) Thirteen phosphorus-dissolving bacteria were isolated and screened in total, and P5 (195.61 mg·L$^{-1}$) had the strongest capacity of phosphorus-solubilized. Five phosphorus-dissolving bacteria were provided with nitrogenase activity, and the highest activity of nitrogenase was P10 and P5 (71.90 C$_2$H$_4$ nmol·mL$^{-1}$·h$^{-1}$ and 71.00 C$_2$H$_4$ nmol·mL$^{-1}$·h$^{-1}$, respectively). Four strains were provided with ACC deaminase activity, and the highest activity of ACC deaminase was P5 and P9, (0.74 μmol·mg$^{-1}$·h$^{-1}$ and 0.54 μmol·mg$^{-1}$·h$^{-1}$, respectively). Most strains could secrete IAA, and three strains of bacteria had a strong secretory ability, which could secrete IAA with a concentration greater than 15 mg·mL$^{-1}$, and P5 was 18.00, P2 was 17.30, P6 was 15.59 (mg·mL$^{-1}$). P5 produced carriers of iron better than others, and the ratio of the diameter of the iron production carrier ring to the diameter of the colony was 1.80, respectively, which was significantly higher than other strains. Combining all kinds of factors, P5 multifunctional phosphorus-dissolving bacteria were screened for eventual further study. (2) Strain P5 was identified as *Burkholderia ubonensis*, based on the colony characteristics, physiological and biochemical tests, 16SrDNA sequence analysis and phylogenetic tree construction. *Conclusion*: P5 has a variety of high-efficiency growth-promoting capabilities, and the ability to produce IAA, ACC deaminase activity and siderophore performance are significantly higher than other strains, which had great potential in the development of microbial fertilizer.

**Keywords:** Chinese fir; soil; phosphorus-dissolving bacteria; nitrogenase activity; promoting characteristics

## 1. Introduction

As one of the important components in the forest soil ecosystem, soil microorganisms play an important role in nutrient conversion and improvement of nutrient utilization [1,2]. In both natural and artificial ecosystems, studies have proved that plants have strong selective effects on soil microbial communities, and the resulting interaction between plants

and microorganisms directly affects plant nutrient absorption and soil nutrient cycling [3,4]. Soil microbial functional bacteria with phosphate-dissolving, nitrogen-fixing and other growth-promoting abilities are the core components of soil microbes because they can improve the soil, form a suitable environment for plant growth and development, increase soil effective nutrient content, and improve soil fertility.

The Chinese fir is an important economical fast-growing timber tree species in China, which plays an important role in the construction and production of national plantations. The latest "China Forest Resources Report (2014–2018)" published in 2020 shows that the number and accumulation of Chinese fir plantations are 21.1 billion plants and 1.079 billion cubic meters, both ranking first in China's artificial arbor forests. However, the continuous planting of pure Chinese fir plantations has caused a serious decline in soil fertility and decreased productivity. Although the construction of mixed forests through tree species combination can effectively alleviate the soil structure and nutrient decline of forest land, the effect of improving fertility by forest stand optimization is limited, and soil N and P are still lacking, seriously restricting sustainable development [5,6]. Many studies have shown that soil available phosphorus content is the main factor limiting the sustainable growth of fir plantations. Because about 75–90% of the phosphorus used in the soil phosphate reacts with iron, aluminum, magnesium, calcium and other metal cations in the soil to turn into insoluble phosphate, resulting in a decrease in the available phosphorus in the soil [7–9]. Compared with nitrogen and potassium fertilizers, phosphate fertilizers are easily fixed by the soil. In the Chinese fir plantation area, the input of phosphate fertilizer is twice the actual absorption capacity of plants, but due to the fixed effect of phosphate fertilizer on the soil, the supply of available phosphorus nutrients in the soil is still insufficient [10]. Relevant research reports point out that with the increasing demand for phosphorus in agriculture and forestry production, the global phosphorus resource reserves will be exhausted in the next 50 to 100 years [11,12]. With the widespread use of phosphate fertilizers and the reduction of phosphorus resource reserves, the cost of phosphate fertilizer production increases, and agricultural and forestry production will face huge challenges [13,14]

Phosphorus plays an important role in the growth of plants. It has the ability to promote root development, accelerate plant growth, optimize water use efficiency and nutrient transport rate. If plants are deficient in phosphorus, their growth is restricted. For example, if phosphorus is deficient, the root system will grow slowly and easily necrosis. Studies have found that there are a large number of bacteria with the ability to dissolve phosphorus in the soil. These bacteria can convert the insoluble phosphate in the soil into water-soluble phosphorus directly used by plants, thereby improving the utilization of soil phosphorus. Phosphorus-dissolving microorganisms have become a current research hotspot because of their energy-saving, low-cost, and wide-ranging sources [15,16]. For example, the phosphate-dissolving bacteria screened by Boroumand [17] can increase the content of nitrogen and available phosphorus in the soil by increasing the activity of soil urease and acid phosphatase. The results of Shahzad et al. [18] indicate that *Rhizobium* phosphate can increase soil dehydrogenase, β-glucosidase, urease activity and soil available phosphorus content. At present, the growth-promoting abilities of growth-promoting bacteria mainly include dissolving insoluble phosphate, fixing nitrogen, producing ACC deaminase, secreting plant growth hormone and siderophore [19]. These abilities can directly or indirectly affect the growth and development of plants. The phosphate-solubilization and nitrogen fixation of the growth-promoting bacteria can indirectly increase the utilization of elements; ACC deaminase can promote plant growth, and plant hormones can regulate the physiological processes of plants themselves. A phosphate-solubilizing bacteria fertilizer is a new type of environmentally friendly biological fertilizer with huge potential for growth promotion. It cannot only improve the absorption of phosphorus by plants and reduce their dependence on traditional fertilizers but also improve soil fertility and optimize the utilization of phosphorus components. Compared with traditional phosphate fertilizer, it is more environmentally friendly and economical [20,21].

Most of the reported rhizosphere growth-promoting bacteria only have a single ability to dissolve phosphorus or potassium. There are few studies on strains with high-efficiency phosphorus-solubilization and nitrogen fixation and other growth-promoting abilities, and their practical applications are greatly limited. For example, independent inoculation of phosphate-dissolving bacteria by Wu et al. significantly increased soil soluble phosphorus content [22]. Studies by Yaish et al. have shown that inoculation of potassium-dissolving bacteria can convert the insoluble potassium in the soil into potassium that can be absorbed and utilized by plants [23]. In order to meet the actual production needs, Wu et al. compounded three kinds of microorganisms with different functions into biological fertilizers. After application, they found that increasing the nitrogen and phosphorus content in the plant and correspondingly improved soil organic matter and total nitrogen [24]. In addition, with the development and demand of modern agriculture, many rhizosphere growth-promoting bacteria have been developed into biological fertilizer preparations and used in agricultural production, while the research and application of multifunctional growth-promoting bacteria in fir soil is rarely reported. Therefore, it is very necessary to isolate and screen multifunctional growth-promoting bacteria from the well-developed rhizosphere soil to promote the sustainable and healthy growth of Chinese firs. In this study, we put forward two hypotheses: (1) High-efficiency phosphate-solubilizing bacteria can be isolated from the soil of a well-grown Chinese fir plantation; (2) part of the phosphate-dissolving bacteria have various growth-promoting properties.

## 2. Materials and Methods

### 2.1. Test Materials

The experimental site is located in the Dagangshan Mountain Forest Farm in Quannan County, Jiangxi Province. The area is located at $114°24'$ E and $27°36'$ N. The geomorphology is a low mountain and hill with an altitude of 220–300 m. The annual average temperature is 17.2 °C, and the annual average precipitation is 1600 mm. It belongs to the subtropical monsoon climate, and the soil type is red soil. The *Cunninghamia lanceolata* plantation is one of the main vegetation types in this area due to continuous afforestation activities and a good ecological environment. For this reason, there are five age-class Chinese fir plantations, including young forest, middle-age forest, near-mature forest, mature forest and over-mature forest. Therefore, it is inferred that the soil microbial organisms in this plantation are diverse and active. Therefore, the Dagangshan Forest Farm is set as an experimental area. In September 2018, seven soil cores 5 cm in diameter and 20 cm deep, more than 1 m away from the tree position, were taken from each plot by the "S-sampling" method. Then, soil cores were mixed well, stored at 4 °C in a portable refrigerator, transported to the laboratory, and removed plant tissues, roots, and rocks before processing.

### 2.2. Preliminary Screening of Phosphate-Dissolving Bacteria

Ten grams of the soil sample was weighed in a triangular flask containing 90 mL of sterile water and shaken on a shaker at 28 °C for 30 min. The soil suspension was continuously diluted 10, 100, 1000 times to obtain the corresponding diluted solution. Of the diluted solution with dilution factors of 100 and 1000, 0.1 mL was taken and spread on the Pikovskava inorganic phosphorus medium (PVK) and incubated at 28 °C for 5–7 days. The colonies with the phosphate ring on the plate were observed, and the colony with the largest phosphate ring was selected. Then, the continuous streaking method was used to further separate it until the isolation and morphology of the colony were uniform for each isolate [25].

The formula of PVK medium used in the experiment was glucose 10 g, glucose 10 g, $Ca_3(PO_4)$ 25 g, $CaCO_3$ 35 g, $(NH_4)_2SO_4$ 0.5 g, NaCl 0.2 g, $MgSO_47H_2O$ 0.1 g, KCl 0.1 g, $MnSO_4$ 0.002 g, $FeSO_47H_2O$ 0.002 g, agar 18 g distilled water 1000 mL, pH 7.0.

### 2.3. Observation Records of Colony Characteristics

Use a sterile inoculation loop to inoculate a single colony on PVK plate medium and put in a 28 °C incubator for 2–7 days. The shape, color, surface state of the colony were observed and recorded.

### 2.4. Promoting Ability Determination

Phosphate-solubilizing bacteria can dissolve not only insoluble phosphate but also have the characteristics of nitrogen fixation, ACC deaminase production, siderophore and IAA secretion to promote plant growth. Therefore, the screening of high-efficiency and multifunctional phosphorus-dissolving bacteria is of great significance to agricultural and forestry production.

#### 2.4.1. Determination of Phosphorus Dissolving Ability

The strain was inoculated into PVK liquid medium and cultured at 28 °C for 180 r/min on a reciprocating shaker for 7 days, and then the pH value of the medium was measured. The culture solution was centrifuged at 8000 RPM for 15 min to remove bacterial cells. Take the supernatant and use a spectrophotometer to determine the content of soluble phosphorus in the culture solution by molybdenum, antimony and scandium colorimetry at a wavelength of 420 nm [26].

#### 2.4.2. Determination of Nitrogenase Activity

The acetylene reduction method measures the activity of nitrogenase [27]. An aliquot of 200 μL fresh culture was inoculated to 20 mL of nutrient broth and incubated overnight at 30 °C. Bacterial growth was collected by centrifugation and was washed twice using sterile water, and resuspended by liquid limited nitrogen culture medium (OD600 = 0.2). The 3 mL suspension was transferred to a 25 mL sterilized serum vial, and 2.4 mL acetylene gas (99.9999%) was driven into the serum bottle and then incubated at 30 °C for 12 h.

#### 2.4.3. 1-Aminocyclopropane-1-Carboxylate (ACC) Deaminase Activity Determination

Determination of ACC deaminase activity by dinitrophenylhydrazine colorimetry [28] using N-free medium for bacteria and minimal medium for actinomycetes containing 0.3 $m \cdot mol \cdot L^{-1}$ ACC (Sigma, Shanghai, China) as the sole nitrogen source. Minimal medium with 0.1% ($w/v$) $NH_4(SO_4)_2$ was used as a positive control, and cultivation without ACC was used as a negative control. After incubation at 28 °C for 7 days for non-actinomycete bacteria and 14 days for actinomycetes, colony growth on N-free medium with addition of ACC indicated ACC deaminase activity.

#### 2.4.4. Indole-3-Acetic Acid (IAA) Production

The Salkowski colorimetric method was used to determine the ability of bacteria to produce IAA [29]. Bacterial isolates were cultured for 3 days in TY broth (without L-tryptophan or supplemented with 500 μg/mL of L-tryptophan) in the dark at 28 °C. Cells were removed from the culture medium by centrifugation at 13,000× $g$ for 10 min; then, 1 mL of the supernatant was mixed vigorously with 2 mL of Salkowski's reagent (4.5 g of $FeCl_3$ per L in 10.8 M $H_2SO_4$). Samples were incubated at room temperature for 30 min, and the IAA production was estimated from the optical density at 600 nm (OD600) by comparison with a standard curve prepared from known concentrations of IAA.

#### 2.4.5. Siderophore Production

The chrome azurol S (CAS) plate method was used to determine the ability of strains to produce iron carriers [30]. Solate was inoculated onto CAS agar, cultured at 28 °C for 2 days, and the positive strain was indicated by an orange halo around the bacterial colony. The ratio ($D/d$) of orange aperture diameter ($D$) to colony diameter ($d$) was determined to determine the iron-producing carrier capacity of the strain.

*2.5. Physiological and Biochemical Tests of Phosphate-Dissolving Bacteria*

Various microorganisms have different enzyme systems, can use different substrates, or use the same substrate to produce different metabolites. Therefore, various physiological and biochemical reactions are used to identify different bacteria.

The conventional physiological and biochemical identification of phosphate-dissolving bacteria is carried out according to the methods in the "Common bacterial system identification manual", which mainly includes Gram stain, glucose hydrolysis test, lactose hydrolysis test, methyl red test, VP test, hydrogen sulfide production test, gelatin liquefaction test, citrate utilization test, malonate utilization test and denitrification test.

*2.6. 16SrDNA*

Taking the screened multifunctional phosphate-dissolving bacteria as the object, a bacterial genomic DNA extraction kit of Beijing Biomed Biotechnology Co., Ltd (Beijing, China). Was used to extract the DNA of the strain. Using the DNA as a template, and using the bacterial universal primer 27F (5′-AGAGTTTGATCCTGGCTCAG-3′) and 1492R (5′-GGTTACCTTGTTACGACTT-3′) PCR amplification, the amplification system was as follows: DNA template 1 μL, primer 27 F 0.5 μL, 1492 R 0.5 μL, 2 × TaqMix12.5 μL, ddH$_2$O 10.5 μL. The PCR procedure is as follows: 93 °C for 3 min, 93 °C for 30 s, 56 °C for 30 s, 72 °C for 2 min, 32 cycles; 72 °C for 7 min. The amplified products were sequenced bidirectionally by BGI. After splicing the measured 16SrDNA sequences in ContigExpress, search in GenBank, EzTaxon, BIGSdb databases, respectively, select the model strains with high homology, construct the phylogenetic tree through MEGA7.0 software, and the confidence level through resampling method (bootstrap = 1000) calculated.

*2.7. Data Processing*

Statistical Product and Service Solutions (SPSS) (IBM, New York, NY, USA) was used to perform a one-way analysis of variance (ANOVA) to evaluate the difference in growth promotion ability between different strains.

**3. Results**

*3.1. Colony Characteristics of Phosphate-Dissolving Bacteria*

A total of 13 strains of phosphorus-solubilizing bacteria were isolated and purified from the soil of the Chinese fir woodland. The morphological characteristics of bacterial colonies generally included shape, color, surface state, gloss and phosphate ring. The morphological characteristics of the 13 isolated phosphorus-dissolving bacteria were observed and recorded. Most of the bacterial colonies were white or light yellow, round, with moist surfaces, and a few colonies were transparent (see Table 1 and Figure 1).

**Table 1.** Colony characteristics of phosphorus-dissolving bacteria on PVK medium.

| Strain no. | P1 | P2 | P3 | P4 | P5 | P6 | P7 | P8 | P9 | P10 | P11 | P12 | P13 |
|---|---|---|---|---|---|---|---|---|---|---|---|---|---|
| Shape | Round | Round | Round | Round | Round | Round | Round | Round | Round | Round | Round | Round | Round |
| Color | Transparent | white | Transparent | Transparent | white | Transparent | white | white | white | Light yellow | white | white | Transparent |
| Surface state | Moist | Moist | Moist | Moist | Moist | Moist | Dry | Moist | Moist | Moist | Moist | Moist | Moist |

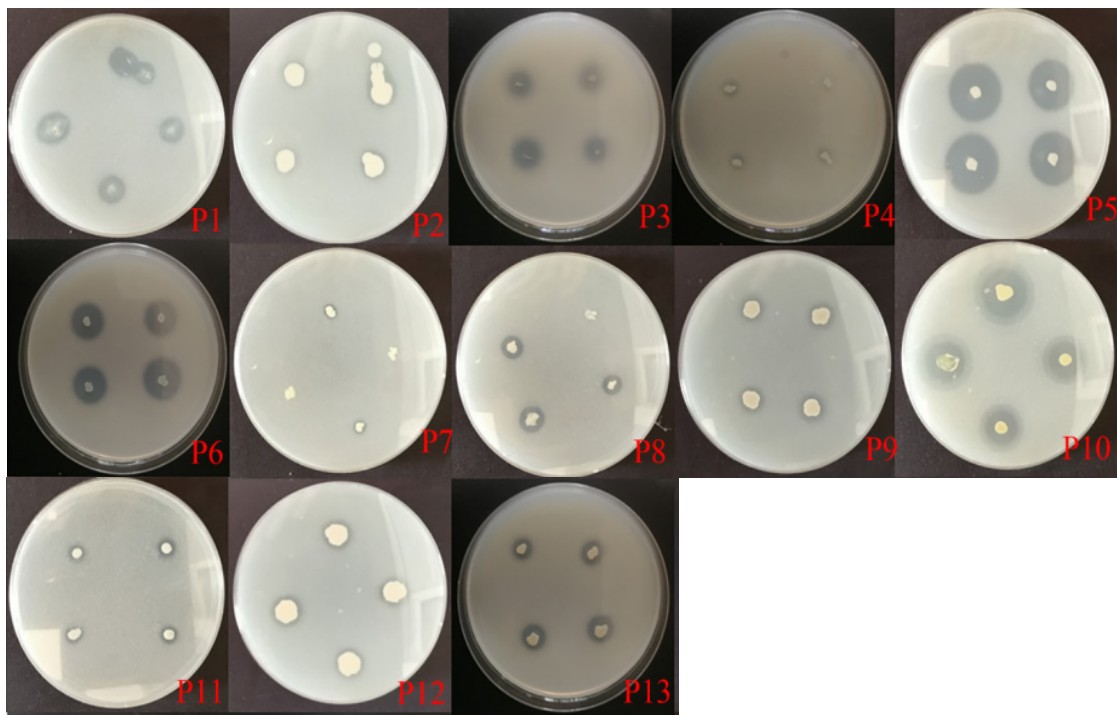

**Figure 1.** Strain growth traits.

*3.2. Research on Growth Promoting Characteristics*

3.2.1. Phosphorus Dissolving Capacity

The strain was inoculated in PVK liquid medium with calcium phosphate as the sole source of phosphorus, and the soluble phosphorus content was measured. The concentration of calcium phosphate was 0.5% (*w*/*v*). Calcium phosphate is slightly soluble in water with a solubility of 25 mg/L. Therefore, the percentage was calculated using the soluble phosphorus content/25. Moreover, the soluble phosphorus content in the culture solution was measured to find that there was a significant difference in the ability of each strain to dissolve phosphorus ($p < 0.05$, Table 2). The soluble phosphorus content was between 21.61–195.61 mg·L$^{-1}$, of which strain P5 had the strongest ability to dissolve calcium phosphate and was significantly higher than other strains; followed by P6 and P2, the soluble content in the culture solution was 56.63 mg·L$^{-1}$, 78.86 mg·L$^{-1}$, 52.58 mg·L$^{-1}$. The weakest phosphorus-dissolving ability were for P1 and P10, which were significantly lower than other strains.

**Table 2.** Determination of phosphorus-dissolving ability.

| Strain No. | Soluble Phosphorus Content (mg L$^{-1}$) | Soluble Phosphorus/25 | Strain No. | Soluble Phosphorus Content (mg L$^{-1}$) | Soluble Phosphorus/25 |
|---|---|---|---|---|---|
| P1 | 21.61 ± 1.82 [e] | 86.44% | P8 | 46.54 ± 3.00 [d] | 145.44% |
| P2 | 52.68 ± 4.86 [bc] | 202.62% | P9 | 36.63 ± 2.85 [d] | 111.00% |
| P3 | 45.67 ± 3.94 [c] | 169.15% | P10 | 24.13 ± 1.81 [e] | 70.97% |
| P4 | 39.62 ± 3.59 [cd] | 141.50% | P11 | 50.77 ± 2.45 [bc] | 145.06% |
| P5 | 195.61 ± 14.9 [a] | 674.52% | P12 | 47.08 ± 1.29 [c] | 130.78% |
| P6 | 56.63 ± 4.29 [b] | 188.77% | P13 | 51.73 ± 1.97 [bc] | 139.81% |
| P7 | 49.54 ± 1.10 [bc] | 159.81% | | | |

Note: significant differences between the thirteen strains of soils were determined using one-way ANOVA followed by Duncan′s multiple range test at <0.05. Different lowercase letters represent significant differences between different strains. The data are shown as the means ± SD (*n* = 3).

### 3.2.2. Nitrogenase Activity

The results of the determination of the nitrogenase activity of phosphorus-solubilizing bacteria are shown in Table 3 ($p < 0.05$). There were 5 strains of phosphate-dissolving bacteria that had nitrogenase activity, accounting for 40% of the tested strains. The nitrogenase activity of the tested strains was in the range of 64.46–71.90 $C_2H_4$ $nmol \cdot mL^{-1} \cdot h^{-1}$, of which the P11 strain had the lowest nitrogenase activity, which was only 7.44 $C_2H_4$ $nmol\ mL^{-1} \cdot h^{-1} \cdot mL^{-1}$, different from the highest nitrogenase activity $h^{-1}$. In general, the nitrogenase activity of these five strains of phosphate-solubilizing bacteria was similar.

**Table 3.** Nitrogenase activity of phosphorus-dissolving bacteria.

| Strain No. | Nitrogenase Activity ($C_2H_4$ nmol mL$^{-1}$ h$^{-1}$) | Strain No. | Nitrogenase Activity ($C_2H_4$ nmol mL$^{-1}$ h$^{-1}$) |
|---|---|---|---|
| P1 | — | P8 | — |
| P2 | 66.55 ± 3.60 [ab] | P9 | — |
| P3 | — | P10 | 71.90 ± 3.88 [ab] |
| P4 | — | P11 | 64.46 ± 3.48 [b] |
| P5 | 71.00 ± 3.84 [ab] | P12 | — |
| P6 | 69.98 ± 3.78 [ab] | P13 | — |
| P7 | — | | |

Note: significant differences between the thirteen strains of soils were determined using one-way ANOVA followed by Duncan′s multiple range test at <0.05. Different lowercase letters represent significant differences between different strains. The data are shown as the means ± SD ($n = 3$).

### 3.2.3. ACC Deaminase Activity

Table 4 ($p < 0.05$) shows the measurement results of the ACC enzyme activity of the 13 test strains. From the table, it can be seen that 4 strains had ACC deaminase activity, and the ACC deaminase activity ranged from 0.20 to 0.74 $\mu mol \cdot mg^{-1} \cdot h^{-1}$. Among them, the P5 strain had the highest ACC deaminase activity, 0.74 $\mu mol \cdot mg^{-1} \cdot h^{-1}$, which was significantly higher than other strains. P9 was next, 0.54 $\mu mol \cdot mg^{-1} \cdot h$. P10 had the lowest ACC deaminase activity, 0.20 $\mu mol \cdot mg^{-1} \cdot h^{-1}$. Overall, the ACC deaminase activity of these four strains differed significantly.

**Table 4.** 1-Aminocyclopropane-1-carboxylic acid (ACC) deaminase of phosphorus-dissolving bacteria.

| Strain No. | ACC Deaminase ($\mu mol$ mg$^{-1}$ h$^{-1}$) | Strain No. | ACC Deaminase ($\mu mol$ mg$^{-1}$ h$^{-1}$) |
|---|---|---|---|
| P1 | — | P8 | — |
| P2 | — | P9 | 0.54 ± 0.02 [b] |
| P3 | — | P10 | 0.20 ± 0.01 [d] |
| P4 | — | P11 | — |
| P5 | 0.74 ± 0.03 [a] | P12 | — |
| P6 | 0.25 ± 0.01 [c] | P13 | — |
| P7 | — | | |

Note: significant differences between the thirteen strains of soils were determined using one-way ANOVA followed by Duncan′s multiple range test at <0.05. Different lowercase letters represent significant differences between different strains. The data are shown as the means ± SD ($n = 3$).

### 3.2.4. Production Capacity of IAA

IAA is the main auxin in plants and participates in many important physiological processes in plants, including cell growth and division, and tissue differentiation. Among the tested strains, 11 strains had the ability to produce IAA, as shown in Table 5 ($p < 0.05$). It can be seen from Table 5 that the strain′s ability to produce IAA was between 10.62 and 18.00 $\mu g \cdot mL^{-1}$, of which strain P5 (18.00 $\mu g \cdot mL^{-1}$) produced the highest amount of IAA. In general, most of the phosphate-dissolving bacteria had the ability to produce IAA, and the ability of each strain to produce IAA was significantly different.

**Table 5.** Indole-3-acetic acid IAA) productions of phosphorus-dissolving bacteria.

| Strain No. | IAA ($\mu$g mL$^{-1}$) | Strain No. | IAA ($\mu$g mL$^{-1}$) |
|---|---|---|---|
| P1 | — | P8 | 14.69 $\pm$ 1.10 [cd] |
| P2 | 17.30 $\pm$ 1.29 [ab] | P9 | 11.93 $\pm$ 0.89 [efgh] |
| P3 | — | P10 | 13.55 $\pm$ 1.01 [cdef] |
| P4 | 14.07 $\pm$ 1.06 [cde] | P11 | 10.62 $\pm$ 0.79 [h] |
| P5 | 18.00 $\pm$ 1.35 [a] | P12 | 11.71 $\pm$ 0.88 [fgh] |
| P6 | 15.59 $\pm$ 1.17 [bc] | P13 | 14.48 $\pm$ 1.1 [cd] |
| P7 | 13.58 $\pm$ 1.01 [cdef] | | |

Note: significant differences between the thirteen strains of soils were determined using one-way ANOVA followed by Duncan's multiple range test at <0.05. Different lowercase letters represent significant differences between different strains. The data are shown as the means $\pm$ SD (*n* = 3).

### 3.2.5. Siderophore Capacity of Phosphate-Dissolving Bacteria

The characteristics of the siderophore of phosphorus-dissolving bacteria were detected by CAS plates. The results are shown in Table 6. It can be seen from Table 6 ($p < 0.05$), the 9 strains could secrete iron carriers, and the size ranges from 1.02 to 1.50. Among them, the ratio of P5, the diameter of the iron carrier circle to the colony diameter was 1.50, which was significantly higher than that of the remaining small strains.

**Table 6.** Ability to produce siderophore of phosphorus-dissolving bacteria.

| Strain No. | Siderophore | Strain No. | Siderophore |
|---|---|---|---|
| P1 | 1.10 $\pm$ 0.07 [cd] | P8 | — |
| P2 | 1.20 $\pm$ 0.08 [c] | P9 | 1.02 $\pm$ 0.07 [def] |
| P3 | — | P10 | 1.03 $\pm$ 0.07 [def] |
| P4 | 1.13 $\pm$ 0.07 [cd] | P11 | — |
| P5 | 1.50 $\pm$ 0.10 [b] | P12 | 1.07 $\pm$ 0.07 [cde] |
| P6 | 1.11 $\pm$ 0.07 [cd] | P13 | — |
| P7 | 1.05 $\pm$ 0.07 [cdef] | | |

Note: significant differences between the thirteen strains of soils were determined using one-way ANOVA followed by Duncan's multiple range test at <0.05. Different lowercase letters represent significant differences between different strains. The data are shown as the means $\pm$ SD (*n* = 3).

### 3.3. Physiological and Biochemical Tests

The selected phosphate-dissolving bacteria were subjected to physiological and biochemical identification. The results are shown in Table 7. Except for P7, the screened strains were all Gram-negative bacteria, glucose hydrolysis test was positive, lactose hydrolysis was negative, methyl red test was only negative for P2 and P4, VP test was positive, hydrogen sulfide test was positive The gelatin liquefaction test was only negative for P2 and P5. The negative for the citrate test was P2, and all other strains were positive. All malonate tests were negative, and only denitrification tests were negative for P1, P7, P10, and P11.

**Table 7.** Physiological and biochemical features of phosphorus-dissolving bacteria.

| Test | Gram Staining | Glucose Hydrolysis | Lactose Hydrolysis | Methyl Red | V-P Test | Hydrogen Sulfide Production | Gelatin Liquefaction | Citrate | Malanate | Denitrification |
|------|------|------|------|------|------|------|------|------|------|------|
| P1 | − | + | + | + | − | − | + | + | − | − |
| P2 | − | + | + | − | − | − | − | − | − | + |
| P3 | − | + | + | + | − | − | + | + | − | + |
| P4 | − | + | + | + | − | − | + | + | − | + |
| P5 | − | + | + | − | − | − | − | + | − | + |
| P6 | − | + | + | + | − | − | + | + | − | + |
| P7 | + | + | + | + | − | − | + | + | − | − |
| P8 | − | + | + | + | − | − | + | + | − | + |
| P9 | − | + | + | + | − | − | + | + | − | + |
| P10 | − | + | + | + | − | − | + | + | − | − |
| P11 | − | + | + | + | − | − | + | + | − | − |
| P12 | − | + | + | + | − | − | + | + | − | + |
| P13 | − | + | + | + | − | − | + | + | − | + |

Note: "+" indicates positive for the test; "−" indicates negative for the test.

*3.4. 16SrDNA Gene Sequence Alignment Analysis*

The 16SrDNA sequence of 13 strains was amplified, and the nucleotide sequence of 16SrDNA of about 1350 bp was obtained. BLAST was performed on GenBank, EzTaxon, and BIGSdb databases, and the results are shown in Table 8. The 16SrDNA sequence comparison results showed that 7 strains of growth-promoting bacteria belonged to 5 genera, among which 5 strains (P1, P3, P8, P12, P13) had high sequence similarity with *Paraburkholderia* (97.7–99.25%). Four strains (P2, P4, P5, P9) had high sequence similarity with *Burkholderia* (96.52–99.92%) and 2 strains (P6, P11) similarity with Erwinia (99.11%, 98.81%). The remaining two strains (P7, P10) had high similarities with *Bacillus*, respectively. *Pantoea*, respectively. The similarity between P7 and *Bacillus aryabhattai* (B8W22) was 100%.

**Table 8.** Identification based on 16SrDNA sequence.

| Strain No. | Type Strain | Identity (%) |
|------|------|------|
| P1 | *Paraburkholderia hiiakae* (JF763857) | 98.78 |
| P2 | *Burkholderia humptydooensis* (MSMB43) | 96.52 |
| P3 | *Paraburkholderia terrae* (NBRC00964) | 98.64 |
| P4 | *Burkholderia arboris* (AM47630) | 99.92 |
| P5 | *Burkholderia ubonensis* (CIP107078) | 98.44 |
| P6 | *Erwinia rhapontici* (ATCC29283) | 99.11 |
| P7 | *Bacillus aryabhattai* (B8W22) | 100 |
| P8 | *Paraburkholderia metrosideri* (DFBP6−1) | 97.7 |
| P9 | *Burkholderia oklahomensis* (C6786) | 98.67 |
| P10 | *Pantoea conspicua* (LMG24534) | 99.63 |
| P11 | *Erwinia billingiae* (CIP106121) | 98.81 |
| P12 | *Paraburkholderia insulsa* (KF733462) | 99.25 |
| P13 | *Paraburkholderia metrosideri* (DFBP6−1) | 98.62 |

**4. Discussion**

Microbial fertilizer is an environmentally friendly fertilizer and represents one of the important directions for the sustainable development of agriculture and forestry in the future. At the same time, there are already some multifunctional strains separated from the rhizosphere of crops at home and abroad and made into biological fertilizer to promote plant growth and development. However, different plant communities and ecological environments and different climate characteristics directly lead to the diversity and differences of microbial communities. In addition, there are many studies on phosphate-solubilizing bacteria, but such studies usually focus only on the phosphate-solubilizing ability of the bacteria while ignoring other studies on the growth-promoting ability. The low soil available phosphorus content is the main limiting factor for the development of

the Chinese fir. Therefore, it is very necessary to isolate and screen multifunctional bacteria from the soil in the specific habitat where the fir grows and make it into a special biological bacterial fertilizer for fir forests [31–33].

Our results support our first hypothesis and partially support our second hypothesis. The growth-promoting strains screened all had the ability to dissolve phosphorus, but in the four indicators of nitrogenase activity, ACC, siderophore, and IAA, only some strains had the corresponding ability. This may have been the result of the strain screening using a PVK phosphate-solubilizing medium. Among the 13 strains, only P5, P6 and P10 had all the growth-promoting properties at the same time, so these three strains were the target multifunctional bacteria studied in this paper.

At present, there are two main ways to dissolve phosphorus by microorganisms. One is acidolysis. Phosphorus-dissolving microorganisms can secrete organic acids during metabolism, which can lower the pH of the medium and dissolve insoluble phosphorus under acidic conditions. The second is enzymatic hydrolysis, that is, phosphorus-dissolving microorganisms that will produce phosphatase, phytase, nuclease, etc. These can chelate with metal ions such as $Ca^{2+}$, $Fe^{3+}$, $Al^{3+}$, and release $PO_4^{3-}$ [34]. The phosphate solubilizing capacity of strain P5 ($195.61$ mg $L^{-1}$) is not only significantly higher than other strains but also higher than most reported phosphate-dissolving bacteria. For example, Kenan Karabo [35] et al. screened the growth-promoting bacteria from the rhizosphere soil of oil palm trees with a phosphorus-solubilization capacity of $55.6$–$168.3$ mg $L^{-1}$. Tao isolated 10 strains of *Bacillus* cereus from the soil, and the conversion value of soluble phosphorus to organic phosphorus was $13.8$–$62.8$ mg·$L^{-1}$. The maximum amount of *Bacillus megaterium* WXD3$-1$ to be isolated from the rhizosphere of lettuce to dissolve $Ca_3(PO_4)_2$ is $93.20$ mg/L [36]. The reason for the huge difference in phosphorus-solubilization capacity may be different ecological, environmental stresses. The first limiting factor for the growth of Chinese firs is the available phosphorus content. The phosphate-solubilizing capacity of the microbial community in the soil of the Chinese fir has been naturally selected and evolved due to the long-term survival of the fittest and iterative, evolutionary process in nature. Compared with fir, the main limiting factor of most plants or plant soils selected by phosphate-solubilizing bacteria is not phosphorus. This directly leads to the huge differences in the phosphorus-solubilization capabilities of the multifunctional bacteria that have been screened. In the optimization study of the culture conditions of the P5 strain [37], the optimal culture conditions of P5 have been verified: the culture time is 72–96 h, the pH is 5–6, the temperature is between 25–30 °C, and the inoculum amount is 1.5%. At the same time, in this study, the growth amount and available phosphorus content of strain P5 were significantly higher than alkaline conditions under acidic conditions, and the influence level of Ph value on the amount of dissolved phosphorus was extremely significant. Therefore, it is speculated that the mechanism of P5 strain's phosphorus-solubilization is acidolysis. Combining Figure 1 and Table 2, it can be found that there are certain limitations in determining the ability of strains to dissolve phosphate by the qualitative determination of the phosphate circle. This may be due to the different diffusion mechanisms of different organic acids. Phosphorus-dissolving circle can be used as a necessary condition for screening phosphorus-dissolving strains, but it is not a sufficient condition. It is not perfect to judge the phosphate-dissolving ability of phosphate-dissolving strains by the size of phosphate-dissolving strain alone [38]. Compared with the qualitative determination of the phosphorus-dissolving ability of phosphorus-dissolving bacteria by the phosphate-dissolving circle, the quantitative determination method of liquid culture is more reasonable and scientific [39,40].

In many previous studies, multifunctional strains that both dissolve phosphorus and fix nitrogen have a synergistic relationship in their growth-promoting ability [41]. However, in this study, the measured values of nitrogenase activity of the three strains were very close, and the difference between the highest P10 and the lowest P6 was only 1.92. This is very special compared with the phenomenon that the P5 strain is significantly higher than the other strains in the other four indicators. Moreover, the nitrogenase activity of P10 is

71.9, which is lower than most nitrogen-fixing bacteria reported in the literature. Nitrogen is the second-most important limiting factor for the growth of Chinese firs. Therefore, in order to increase the soil nitrogen content and increase the C/N ratio, thereby increasing the utilization of phosphorus, a large amount of nitrogen fertilizer is usually applied to Chinese fir plantations. According to related studies [42,43], an excessive supply of nitrogen fertilizer will cause a significant decrease in nitrogenase activity. These results indicate that environmental stress has a direct and important impact on the growth-promoting properties of soil microorganisms, rather than a synergistic relationship. This provides a reference basis for screening functional strains with specific growth-promoting properties. ACC deaminase is a characteristic enzyme shared by many functional growth-promoting bacteria [44]. It plays an important role in the resistance of plants to high-temperature, high salt, cold, and disease. The study found that four strains have ACC deaminase activity; the activity range is 0.2–0.74 $\mu$mol mg$^{-1}$ h$^{-1}$. ACC deaminase is an intracellular enzyme that inhibits ethylene biosynthesis. When microorganisms with ACC deaminase activity are inoculated in the rhizosphere of plants, they can reduce the excessive ethylene concentration produced by plant nutrient stress and promote plant growth. IAA is an important plant hormone that participates in many important physiological processes in plants, including cell growth and division, tissue differentiation, and promotes the growth of plant roots and buds. Most phosphate-solubilizing bacteria can promote plant growth by secreting IAA. This study found that 11 strains can produce IAA, and the yield of IAA is relatively high. At present, IAA production mainly adopts chemical methods. Indole, formaldehyde and potassium cyanide react at 150 °C and 0.9–1.0 MPa to produce 3−indole acetonitrile, which is then hydrolyzed under the action of potassium hydroxide, or from indole and potassium cyanide. It is obtained by the reaction of glycolic acid [45,46], which requires high-temperature and high-pressure production conditions, and produces a large amount of hazardous waste liquid. If it can be produced by biological fermentation, it cannot only reduce the cost but also will not cause pollution to the environment.

Siderophores are small organic molecular compounds produced by microorganisms under iron-deficiency conditions. Their role is to obtain insoluble iron compounds from the environment and convert them into a form that can be absorbed by plants, and promote their growth by enhancing the absorption of iron by plants [47]. The literature has shown that both Gram-negative bacteria and Gram-positive bacteria can produce siderophores, which is consistent with the results of this study [48]. The diameter ratio of the siderophore ring produced by P5 is 1.50, which is quite different from the existing strains. For example, the diameter ratio of seven *Pseudomonas* siderophore-producing strains selected by Chen Shaoxing [49] is between 1.50 and 12.00. It may be that the siderophore production capacity of strains is mainly related to the genetic characteristics of the strains, the types of siderophores secreted by different strains are different, and the types of siderophores are related to the strength of the iron chelation reaction in the soil.

The study found that most of the phosphate solubilizing bacteria screened out were Gram-negative bacteria, which was consistent with Kirgiz [31] and Tang A [50] in their research on the isolation of endophytes from rhizosphere soil, indicating that Gram-negative bacteria strains have the characteristics of dissolving insoluble phosphate. Therefore, 16SrDNA gene sequence sequencing to identify bacteria has been widely used in the biological field. In this paper, combining the characteristics of the colony, physiological and biochemical tests, 16SrDNA sequence comparison, the strains are identified to species. Among the phosphate-solubilizing bacteria isolated in this study, *Burkholderia* parabens had the largest number, followed by *Burkholderia* and *Pseudomonas*. The currently known phosphate-solubilizing bacteria mainly include *Burkholderia*, *Bacillus*, *Pseudomonas*, Erwinia, *Klebsiella*, among which *Pseudomonas*, *Burkholderia* and there are many research reports on *Bacillus*.

## 5. Conclusions

The production cost of bio-fertilizer made from multifunctional growth-promoting bacteria is low. After application, it cannot only increase the content of available phosphorus in the soil, promote plant growth but also improve saline–alkali land and soil structure, which is of great significance for maintaining the balance of the ecological environment. Among the 13 strains obtained from the soil of the Chinese fir forest, 13 strains with nitrogen fixation ability were screened through various growth promotion assays and 16S rDNA identification. Of all the growth-promoting properties such as the vector and IAA, only three strains had the ability to dissolve phosphorus, nitrogenase activity, ACC and iron. The remaining nine strains had only part of the growth-promoting ability. These three strains were P5, P6 and P10, respectively, and had good growth-promoting potential. Our research provides more and better choices for multifunctional bacteria as well as provides raw materials for the development and promotion of microbial fertilizer. At the same time, the multifunctional growth-promoting ability of P5, such as phosphorus-solubilization, nitrogen fixation, ACC deaminase production, IAA production and siderophore production was significantly higher than other strains, and the phosphorus-solubilization ability was higher than most of the reported strains. Therefore, a bio-fertilizer made from P5 has great economic value. In the follow-up work, on one hand, the phosphate-solubilization mechanism of the strain should be studied in-depth, and on the other hand, the growth-promoting effect and colonization effect of the strain should be further studied under natural conditions.

**Author Contributions:** G.Z. and Y.W. designed the study and drafted the article. J.C. designed the separation and identification work. Y.D. and L.H. helped to collate the experimental data and assessed the presented data. Supervision: R.J. All authors helped to edit the research paper. The author(s) read and approved the final manuscript. All authors have read and agreed to the published version of the manuscript.

**Funding:** This work is supported by the State Key Research and Development Program of China (2016YFD0600300), the Special Fund for Basic Scientific Research Business of Central Public Research Institutes (CAFYBB2018SY001).

**Data Availability Statement:** No new data were created or analyzed in this study. Data sharing is not applicable to this article.

**Conflicts of Interest:** The authors declare no conflict of interest.

## Abbreviations

| | |
|---|---|
| ACC | 1-Aminocyclopropane-1-carboxylate |
| CAS | Chrome azurol s |
| d | Colony diameter |
| IAA | Indole-3-acetic acid |
| D | Orange aperture diameter |
| PVK | Pikovskava inorganic phosphorus medium |

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
