# Peer review of "Screening, Identification and Growth-Promotion Products of Multifunctional Bacteria in a Chinese Fir Plantation"

_forests, doi:10.3390/f12020120_

Round 1
Reviewer 1 Report
Review of
"Screening, Identification and Growth Promoting Characteristics of Multifunctional Bacteria in Chinese Fir Plantation"
by
Guangyu Zhao, Yihui Weia, Jiaqi Chena, Yuhong Dong
The manuscript concerns an important topic for the development of forest ecosystems, which is increasing the access of trees to phosphorus contained in the soil.
The manuscript needs to be revised in order to publish.
General comments:
- The title of the manuscript misleads the readers - it indicates that the results of the identification of bacteria participating in the uptake of phosphorus from the ground will be presented (in fact, only one has been indicated), and that the conditions for their growth promoting will be indicated - bacteria or firs ? (this section is very poorly described in the manuscript)
- It is necessary to clearly indicate what is the novelty of the research presented in the manuscript.
- A lot of abbreviations are used in the manuscript without their explanation - the explanations should be supplemented in the text or a separate section with the abbreviations used should be added
- The introduction focuses on local issues in China - it is necessary to expand the paragraph on similar research on a global scale
- The description of the research methodology requires intensive improvement. Especially:
- 2.1: provide detailed information on the selected area, e.g. why it was selected, what is it special, from which area 7 soil samples were taken;
- 2.1.1: the choice of PVK solid medium should be justified
- which was used to dilute the soil suspension
- 2.4 and 2.5: it should justify the scope of the research (what was their task in the context of the purpose of the study), and not only refer to their methodology
- 2.7: what method of variance analysis was used for the test results?
- Results and analysis:
- if in the manuscript uses terms significantly higher or lower, the results of the statistical analysis of the results should be presented (e.g. Anova) (e.g. line 178, 196, 204)
- 3.1: There is no explanation on what basis it was found that 13 strains of bacteria were isolated and why it is indicated that 7 of them are phosphorolytic bacteria - the data in Table 1 cannot be the basis for such conclusions
- 3.2.1: what was the concentration of calcium phosphate in the medium, i.e. what is the theoretical maximum concentration of the dissolved fraction
- Table 2: data should be supplemented with the efficiency of phosphate release;
- Table 2-6: what do the symbols after the values mean (e.g. ab)
- 3.4: why the SrDNA analysis results for the remaining bacterial strains were not presented - this significantly limits the possibility of their identification, and at the same time the entry “The phylogenetic tree showed that 19 strains had significant similarity to the gene series of model strains” (line 299-300) ) indicates that such research has been performed
- Discussion
- the discussion is limited to a simple comparison of the obtained numerical values with the values presented in the literature
- it is necessary to present the expected mechanism of calcium phosphate dissolution by the analyzed bacterial strains, including the comparison to that used by fungi
- there is no discussion of the obtained results of the activity of the analyzed bacterial strains in the context of the possibility of their application to other areas, as well as in the context of literature reports
- what are the growing conditions for Multifunctional Bacteria?
- the discussion on 16SrDNA is pointless in the absence of a complete set of results
- Conclusions: the conclusions contain a simple summary of the results, without indicating their importance for the development of knowledge in the analyzed field of science and without indicating the novelty. Practical possibilities of applying the obtained research results should also be indicated.
Detailed comments:
- line 135: complete the year in the cited literature (this error is repeated often in the manuscript)
- Line 142: improve on 2.6 16SrDNA
- Line 226: improve on 3.4 16SrDNA gene sequence alignment analysis
Author Response
All modifications are in the attachment

Reviewer 2 Report
Forests 1023046
Screening, Identification and Growth Promoting Characteristics of Multifunctional Bacteria in Chinese fir Plantation
Review 201217 (same as due date)
Zhao et al. screened and isolated several species of bacteria that are potentially useful for developing microbial fertilizers, in which live microorganisms improve nutrient supply to plant via their metabolic abilities such as solubilizing phosphorus. It appears likely that they were successful, but unfortunately the severe problems of sentence structure, formatting, and adherence to presentation customs make it difficult to evaluate the manuscript. The methods are poorly described and leave the reader asking many questions about what, exactly, was done in the experiment and analyses. The results are disjointed and it is very difficult to extract meaningful comparisons among the bacterial strains. The discussion consists largely of text that would fit better in the results (and the resuls is likewise full of text better found in the methods), with few syntheses of the results or implications for future work beyond statements about the performance of the best-performing bacterial strain.
I believe this study includes some very interesting and useful results, but in its current form I cannot fully evaluate it. I suggest the authors heavily revise the writing, and pay much more attention to the formatting and overall appearance of the manuscript as well as important components such as the tables and figures.
Detailed comments follow:
Introduction
LN 79: “but there are few studies on forest trees that are close to blank.” Is there another word that should replace “blank” here?
LN 83: “More environmentally friendly than” than what? More environmentally friendly than the usual chemical fertilizers?
LN 92-93: The final sentence here starting with “The use of” is unnecessary and reads like a sub-section heading.
Materials and methods
LN 97: the latitude and longitude appear to be reversed.
LN 103: this is unclear. Was the mixed soil sample frozen for transport to the lab? At what temperature? Was it still frozen when it arrived at the lab?
LN 104-108: What is this? There are no complete sentences here, just a list of components. What medium? The growth medium?
LN 110: this is written as an instruction. Please do not tell me what to do, tell me what you did. LN 113—115 has the same problem. This kind of instruction rather than description appears throughout the methods section.
LN 111: The dilutions are not clear. What does “diluted by 10-2” mean? A factor of 8? A factor of 100?
LN 118: What else was observed and recorded beside shape, color, and “surface state” (whatever that is, unclear)? The use of “etc.” here implies a long list of other characteristics. What were they?
LN 119-135: Section 2.4. All of the sub-sub-sections here could be better combined into a single paragraph instead of a series of five sub-sub-sections each of one sentence.
LN 136-141: This sub-section is vague and meaningless. Do you have a citation for the document titled “Common Bacterial System Identification Manual”? And what does “which mainly includes” mean – are there other tests that you conducted but do not wish to tell us about?
LN 147: I suspect you did not actually use an entire litre of DNA template. Please check your units.
LN 156: I appreciate the aesthetic efforts, but beautification of scientific figures is not normally necessary.
Results and analysis
- why “and analysis” ?
LN 161: again, the use of “etc.” implies other details that might be important, but you have chosen not to describe them.
LN 180: Table 2. What do the letters indicate? Statistically significant differences?
Tables 2 through 6 should be combined into a single, properly-formatted table to summarise the results.
Parts of the Results text would be more appropriate in the Methods section.
Discussion
LN 238: “The effectiveness of phosphorus on plant roots.” This is an incomplete sentence that reads like a sub-section heading. Similar incomplete, heading-like sentences appear throughout the Discussion.
Author Response
All modifications are in the attachment

Round 2
Reviewer 1 Report
Comments on the revised manuscript in the attached file. Items requiring further response are underlined

Author Response
All replies are attached

Reviewer 2 Report
Screening, Identification and Growth Promotion Products of Multifunctional Bacteria in Chinese Fir Plantation.
Guangyu Zhao et al.
Version 3
Forests
First Pass 210106
Abstract
- Introduction
LN 49: “Secondly, it is nitrogen content.” this is not a complete sentence.
LN 51-52: “Studies have shown” to “effectiveness of phosphorus” – this sentence is very unclear. If Chinese fir leaves have a N:P ratio below the plant growth threshold of 14, how can the leaves have grown in the first place? What do you mean by “plant growth threshold” ? Is growth completely prevented below this threshold?
LN 56: “Such as poor root development and easy aging.” again, a problem sentence. And what is meant by “easy aging”?
The use of the word “fix” (and “fixation” and so forth) in this paragraph is confusing. Does soil fix phosphorous, or is phosphorus fixed by soil microorganisms such as bacteria? Is soil itself fixed, as implied on LN 62?
LN 64: the switch to discussion global phosphorus supply should be a separate paragraph.
LN 86: “development and development” ??
Sorry, the problems with English language use in this manuscript are still too severe to allow detailed review. Furthermore, while both the English use and the overall structure of the manuscript is improved compared to the previous version I reviewed (file labelled V1) both aspects are far from acceptable. Many inexplicable changes in line spacing, font, font size, and other aspects of formatting are present throughout the manuscript, and it is clear that little attention has been paid to proofreading. Many of the problems I found in the previous version have not been addressed, such as the text within Tables. Figure 1, which is paired with Table 1, is a good addition, though it needs a proper figure caption. These problems create doubt in my mind about the intended meaning of sentences that appear to describe methodological procedures, experimental results, and the nuances of the Discussion section. In short, it is too unclear to evaluate.
Again, I would like to aknowledge that the authors have made significant improvements, but this manuscript is effectively incomplete and impossible to fairly review until the overall quality of the writing and presentation is raised further.
Author Response
The modified content is in the attachment
